# Bridging 3D Editing and Geometry-Consistent Paired Dataset Creation for 2D Nighttime-to-Daytime Translation

Xiao Cao    Yuyang Zhao    Robby T. Tan    Zhiyong Huang

National University of Singapore

{xiaocao, yuyang.zhao}@u.nus.edu    {robby.tan, dcshuang}@nus.edu.sg

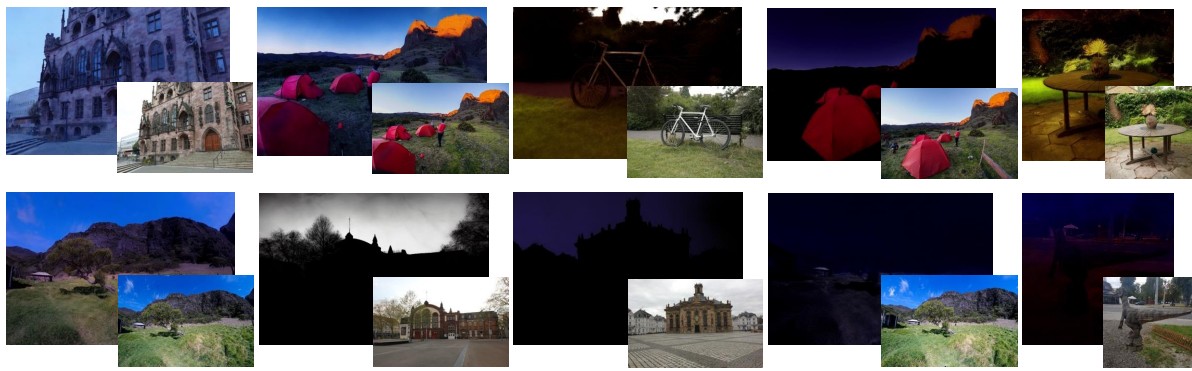

Figure 1. Geometry-consistent paired nighttime-to-daytime translation dataset generated by our proposed 3D editing based pipeline. Each scene contains image pairs under various nighttime light conditions.

## Abstract

*Creating paired nighttime-to-daytime translation datasets remains a challenging and impractical task, as keeping every object static at different times is impossible. While 2D generative models can synthesize paired data for appearance and style translation, they often fail to maintain geometric consistency. In this paper, we propose a novel paired synthetic dataset creation pipeline that leverages 3D editing techniques to convert daytime 3D datasets into nighttime degraded scenes, generating geometrically consistent high-quality image pairs. Through this approach, we construct the first paired synthetic dataset for nighttime-to-daytime translation with geometric consistency. The synthesized data pairs can effectively enhance nighttime-to-daytime editing performance of various 2D generative models both qualitatively and quantitatively, demonstrating the advantages of using 3D editing for paired synthetic visual dataset generation. Code and Dataset are available at github.com/massyzs/3DEdting4Translation.git.*

## 1. Introduction

Nighttime-daytime image translation benefits many real-world applications, such as robust autonomous driving. The success of image translation cannot apart from the paired images. However, obtaining geometry-consistent real-world paired images of different lighting conditions is challenging due to interference from dynamic factors (e.g., vehicles, and natural elements).

Existing real-world nighttime-to-daytime datasets [10, 15, 19] mainly focus on semantic learning. The data pairs are commonly collected by taking videos of the same scenario in daytime and nighttime separately. Therefore, dynamic object interference may exist during the collection process and thus impairs the geometry consistency and incurs additional effort on tackling the unpaired data problem [8, 24, 28, 29]. In view of the limitation of paired data, previous works [1, 11, 26, 30] mainly focus on developing techniques to generate consistent results learned from the inconsistent pairs. While these works achieve remarkable performance, using paired data has the potential to further improve their image transfer ability.

Motivated by the data limitation, we investigate the geometry-consistent nighttime-daytime data generation in this paper. Intuitively, advanced image editing tech-

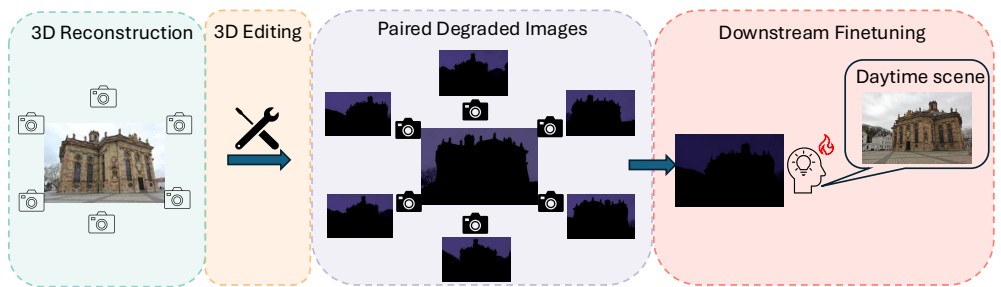

Figure 2. Paired dataset creation pipeline: Our pipeline first applies 3D editing on reconstructed 3D scenes to adjust light conditions, generating geometry-consistent image pairs. These paired images are subsequently used to fine-tune 2D generative models for downstream nighttime-to-daytime translation tasks.

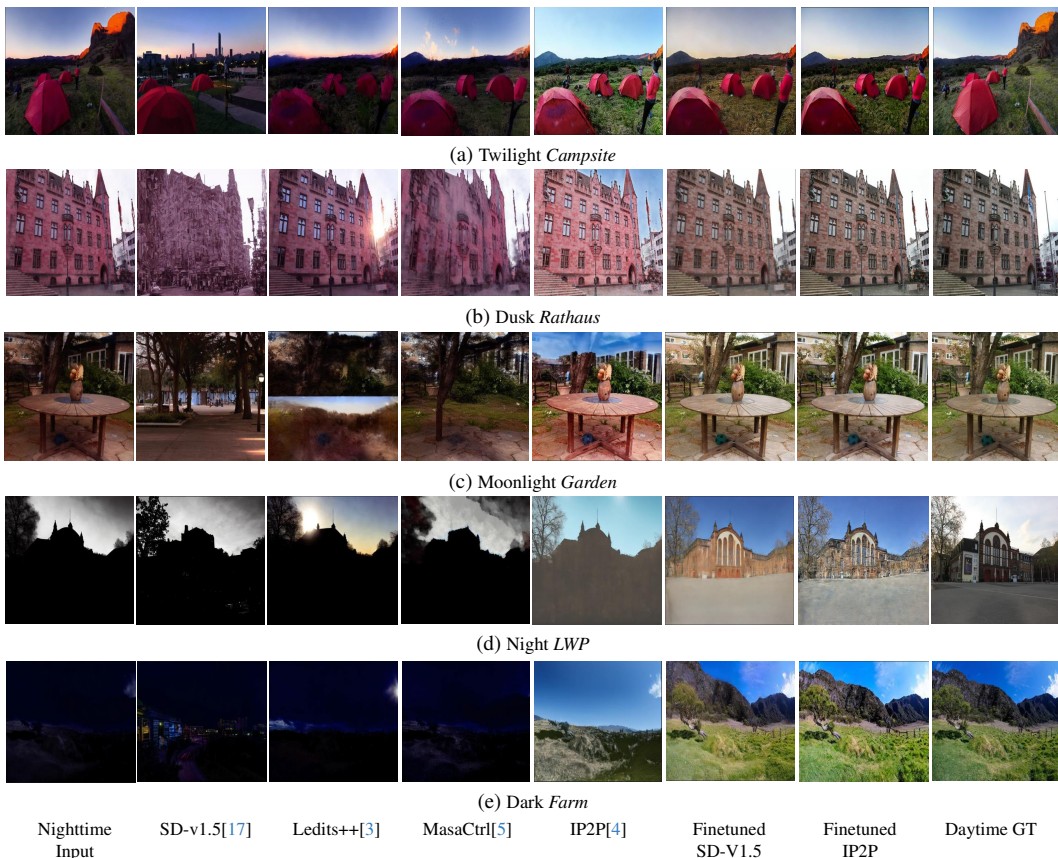

(a) Twilight *Campsite*

(b) Dusk *Rathaus*

(c) Moonlight *Garden*

(d) Night *LWP*

(e) Dark *Farm*

| Nighttime Input | SD-v1.5[17] | Ledits++[3] | MasaCtrl[5] | IP2P[4] | Finetuned SD-V1.5 | Finetuned IP2P | Daytime GT |

Figure 3. Qualitative results of nighttime-to-daytime translation obtained on our paired synthetic testset. Scene names from the original 3D dataset are denoted in *italic*.

niques [3–5] can help generate paired data by editing the image style. Therefore, we first evaluate the performance of image editing baselines (Fig. 3 and Fig. 4) and observe that none of the methods can generate geometry consistent results in daytime style.

To this end, we propose a 3D editing based paired synthetic dataset generation pipeline. Our pipeline can effectively convert existing daytime images from 3D datasets into geometrically consistent images under various night lighting conditions, resulting in 3,812 image pairs. The generated pairs are used to finetune the 2D generative models [4, 17], significantly improving the translation results.

Our contributions can be summarized as:
- We propose a nighttime-to-daytime paired data generation pipeline based on 3D editing.
- We collect multiview 3D scene data and convert them from daytime to various nighttime lighting conditions, forming a dataset with 3,812 image pairs.

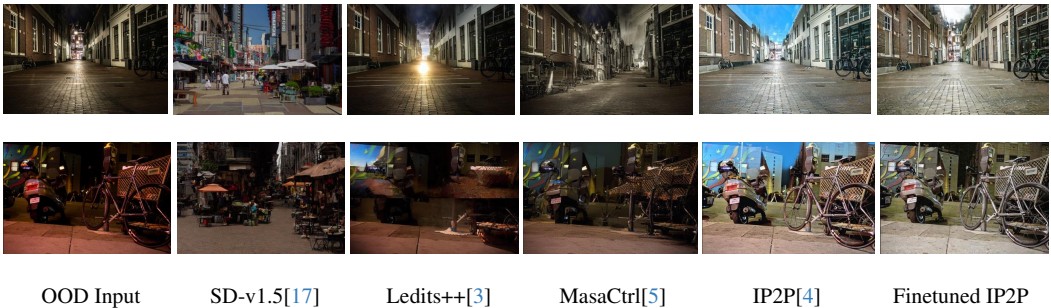

| OOD Input | SD-v1.5[17] | Ledits++[3] | MasaCtrl[5] | IP2P[4] | Finetuned IP2P |

Figure 4. Qualitative results of nighttime-to-daytime translation obtained on Exdark [13] dataset.

- We evaluate the performance of 2D image editing methods and demonstrate how our paired synthetic dataset effectively enhances the nighttime-to-daytime translation capability of 2D generative models.

## 2. Related Works

### 2.1. Nighttime-to-Daytime Image Translation

Many 2D generative models have been proposed to address the nighttime-to-daytime translation task. BBDM [12] employs a bidirectional diffusion process for image translation between different domains. Plug-n-Play [20] integrates source image features to preserve structure without additional training. To accelerate the translation process, One-step Diffusion [25] leverages two diffusion models adversarially, producing more realistic results in a single inference step. Instruct-Pix2Pix [4] finetunes the vanilla Stable Diffusion model with instruction-based prompts, enhancing its suitability for image translation. However, existing models struggle to maintain geometric consistency when performing nighttime-to-daytime translation. Therefore, there is an urgent need for a geometry-consistent paired nighttime-to-daytime translation dataset.

### 2.2. 3D editing

Recent 3D editing methods [6, 7, 9, 21, 23] leverage the capabilities of 2D generative models to perform geometry-aware and view-consistent 3D scene editing. Specifically, Instruct-NeRF2NeRF [9] and Instruct-GS2GS [21] ensure view consistency through iterative editing combined with 3D scene fine-tuning. GaussCtrl [23] introduces a cross-attention mechanism that integrates 3D geometry and multi-view image information to simultaneously maintain geometric and texture consistency. Meanwhile, DGE [7] incorporates epipolar geometry constraints to preserve consistency while enabling editing with significant texture semantic modifications. In this work, we leverage a naive 3D editing baseline instruct-NeRF2NeRF in our proposed pipeline to show its effectiveness for constructing paired datasets.

## 3. Dataset

### 3.1. Dataset Creation Pipeline

Our paired image translation generation pipeline contains 4 parts: (1) 3D reconstruction, (2) 3D editing, (3) rendering degraded images, and (4) generative models downstream finetuning. Two 3D datasets of different scene structures are used, including 360-degree 3D scene [2], and 3D relighting [18]. All of these data are daytime and our aim is to convert them to nighttime with various lighting conditions. We first conduct 3D reconstruction with NeRF [14], which compresses a 3D scene into a neural network and can render arbitrary views. After reconstruction, 3D editing is applied following Instruct-NeRF2NeRF [9]. Specifically, an image is rendered from the trained NeRF and edited with a pre-trained IP2P [4]. Then the edited image is used to fine-tune the NeRF. This process will gradually change the style of NeRF-rendered images while maintaining 3D geometry consistency. For the editing instructions, we set "Dark night", "Night time", "Twilight", "Dusk" and "Moonlight" as style prompts, and use "Turn this into a {*style*} scene" as the prompt template for 3D editing. After 3D editing stage, the scenes with low style alignment score are filtered out.

Our final goal is to synthesize image pairs that can benefit the nighttime-to-daytime translation. Therefore, to avoid the blur due to the lack of observation in 3D reconstruction, we only render images with viewpoints corresponding to existing daytime ground-truth images, *i.e.*, views originally used for training the 3D reconstruction, thereby ensuring geometry-consistent image pairs. The synthesized image pairs can effectively improve the performance of the specific image translation and they are applicable to various base models. In this paper, we finetune Stable Diffusion v1.5 [17] and Instruct-Pix2Pix [4] to demonstrate the effectiveness of the synthetic data.

### 3.2. Dataset Statistic

We present statistics of our synthetic paired nighttime-to-daytime translation dataset in Tab. 2. After filtering out low-quality scenes, the final dataset comprises 8 scenes under 5

| Method | Twilight | | | Dusk | | | Moonlit | | |
|---|---|---|---|---|---|---|---|---|---|
| | CLIP ↑ | SSIM ↑ | LPIPS ↓ | CLIP ↑ | SSIM ↑ | LPIPS ↓ | CLIP ↑ | SSIM ↑ | LPIPS ↓ |
| SD-v1.5[17] | 0.5786 | 0.2684 | 0.6475 | 0.5769 | 0.2404 | 0.6523 | 0.5782 | 0.2374 | 0.6614 |
| IP2P[4] | 0.8389 | 0.4290 | 0.3670 | 0.8126 | 0.4335 | 0.3725 | 0.8594 | 0.4404 | 0.3413 |
| Ledits++[3] | 0.7463 | 0.4097 | 0.5457 | 0.7276 | 0.3726 | 0.5375 | 0.7160 | 0.3893 | 0.5623 |
| MasaCtrl[5] | 0.7789 | 0.3996 | 0.4939 | 0.7467 | 0.3572 | 0.5039 | 0.7757 | 0.3786 | 0.5051 |
| Finetuned SD-v1.5 | 0.9067 | 0.4554 | 0.3048 | 0.9016 | 0.4535 | 0.2929 | 0.9239 | 0.4579 | 0.2684 |
| Finetuned IP2P | **0.9335** | **0.5017** | **0.2332** | **0.9322** | **0.4879** | **0.2412** | **0.9322** | **0.4855** | **0.2369** |

| Method | Night | | | Dark Night | | | Average | | |
|---|---|---|---|---|---|---|---|---|---|
| | CLIP ↑ | SSIM ↑ | LPIPS ↓ | CLIP ↑ | SSIM ↑ | LPIPS ↓ | CLIP ↑ | SSIM ↑ | LPIPS ↓ |
| SD-v1.5[17] | 0.5694 | 0.2086 | 0.7498 | 0.5672 | 0.0812 | 0.8072 | 0.5746 | 0.2213 | 0.6941 |
| IP2P[4] | 0.6703 | 0.3142 | 0.5970 | 0.6638 | 0.3626 | 0.5921 | 0.7752 | 0.3958 | 0.4459 |
| Ledits++[3] | 0.6294 | 0.2414 | 0.7890 | 0.7276 | 0.3726 | 0.5375 | 0.6985 | 0.3205 | 0.6350 |
| MasaCtrl[5] | 0.6247 | 0.2326 | 0.7618 | 0.6685 | 0.0855 | 0.7894 | 0.7199 | 0.3104 | 0.5978 |
| Finetuned SD-v1.5 | **0.8271** | **0.4047** | 0.4429 | 0.9053 | 0.4157 | 0.3083 | 0.8882 | 0.4383 | 0.3311 |
| Finetuned IP2P | 0.8263 | 0.3898 | **0.3948** | **0.9154** | **0.4193** | **0.3000** | **0.9036** | **0.4582** | **0.2841** |

Table 1. Quantitative results of baselines and finetuned generative models on synthetic nighttime-to-daytime translation dataset under different light conditions. **Bold** numbers refers to the best performance and underlined numbers refers to the second best performance.

distinct light conditions, effectively covering diverse real-world scenarios with varying degrees of nighttime degradation, ranging from easier conditions (i.e., twilight) to more challenging conditions (i.e., dark night). We visualize representative examples from the proposed dataset in Fig. 1. From top-left (Twilight) to bottom-right (Dark night), scene visibility gradually decreases, correspondingly increasing the difficulty of nighttime-to-daytime translation.

## 4. Experiment

### 4.1. Quantitative Results

We conduct quantitative comparisons for SD-v1.5 [17], Ledits++ [3], MasaCtrl [5], IP2P [4], and finetune SD-v1.5 and IP2P on our proposed dataset. Evaluated by CLIP [16], SSIM [22] and LPIPS [27] scores, finetuned baselines outperform other baselines significantly under all scenarios with various light conditions. This demonstrates that our paired dataset can help enhance 2D generative models' image transfer ability. Due to the unpaired setting of real-world nighttime datasets, we only conduct qualitative comparisons in following section.

### 4.2. Qualitative Results

**In-distribution Dataset.** We present qualitative results obtained on our dataset in Fig.3. Under all tested conditions, finetuned methods consistently outperform baseline models. As shown in Fig.3a, 3b, and 3c, baseline methods such as SD-v1.5, Ledits++, and MassCtrl introduce artifacts, and they fail significantly under extremely dark nighttime conditions (Fig.3d and 3e). In contrast, the fine-tuned methods effectively translate nighttime images into realistic daytime images without artifacts.

| Split\Scene | Night garden | Night lk2 | Night lwp | Night rathaus | Night stjacob | Night dinosaur | Night bicycle | Dark night farm | Dark night campsite | Dusk campsite | Dusk farm |
|---|---|---|---|---|---|---|---|---|---|---|---|
| Train | 167 | 157 | 94 | 135 | 135 | 135 | 174 | 242 | 157 | 157 | 242 |
| Test | 18 | 17 | 10 | 15 | 15 | 15 | 19 | 42 | 17 | 17 | 42 |

| Split\Scene | Dusk garden | Dusk rathaus | Moonlit campsite | Moonlit farm | Moonlit garden | Moonlit rathaus | Twilight campsite | Twilight farm | Twilight garden | Twilight rathaus | Total |
|---|---|---|---|---|---|---|---|---|---|---|---|
| Train | 167 | 135 | 157 | 225 | 167 | 135 | 157 | 225 | 181 | 135 | 3410 |
| Test | 18 | 15 | 17 | 25 | 18 | 15 | 17 | 25 | 33 | 15 | 402 |

Table 2. Statistic of our paired synthetic nighttime-to-daytime image translation dataset. Each scene contains different nighttime light conditions.

**Out-of-distribution Dataset.** To further verify whether this performance improvement originates from overfitting to specific scenes or genuine nighttime-to-daytime translation capability, we further evaluate methods on the unpaired Exdark dataset [13] (Fig. 4). Across the baselines, only IP2P partially translate input images into daytime while artifacts are introduced. The blue sky is translated into blur roof in the first scene and the building is merged with blue sky in the second scene. Instead, the finetuned model achieves the most satisfying results with realistic sky and structure-preserved buildings. Such results further demonstrate our proposed paired synthetic dataset effectively enhances the nighttime-to-daytime translation capability of 2D generative models.

## 5. Conclusion

Creating geometry-consistent paired datasets is a consuming and challenging task, particularly in the nighttime-to-daytime translation domain. In this work, we propose an effective 3D editing based generating paired datasets pipeline and establish the first geometry-consistent night-to-dayt paired dataset. Experimental results demonstrate the effectiveness of our paired data generation pipeline and highlight out synthetic paired dataset's utility.

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
