# OpenReview forum: "Bridging 3D Editing and Geometry-Consistent Paired Dataset Creation for 2D Nighttime-to-Daytime Translation"
_thecvf.com/CVPR/2025/Workshop/SyntaGen — SyntaGen 2025 Poster_

### Official Review · Reviewer_EMf2 · 2025-03-27
**The dataset and the proposed research problem are interesting, but a comparison to other similar datasets should be added.**

**Rating:** 6
**Confidence:** 3

**Review:**

Summary

Existing methods of nighttime-to-daytime image translation typically rely on finetuning real day-night photo-paired datasets captured at different times. However, such datasets often introduce inconsistencies, such as object changes, due to the time interval between captures. Instead of training on these inconsistent pairs, this paper proposes a synthetic day-night paired dataset with geometric consistency. The dataset is created by leveraging 3D scene datasets and existing 3D editing methods to modify only the neural representation's color properties to simulate different times of day while preserving structural consistency. These edited 3D scenes are then rendered to generate synthetic day-night image pairs. This paper shows that finetuning image editing models on this dataset significantly improves day-night translation performance compared to vanilla models.

Strength
- The problem setup is interesting and aligns well with the workshop theme.
- The paper is well-structured and easy to follow, with a clear explanation of dataset generation.
- The quantitative results are strong, showing that finetuning on the proposed dataset significantly improves translation performance across all time conditions (e.g., Twilight, Dusk, Moonlit, etc.).
- The qualitative results also look promising, as they show geometric consistency even when tested on real images.

Weakness

- The paper lacks a comparison with image editing finetuned on existing nighttime-to-daytime datasets. Without such a comparison, it is unclear whether the proposed dataset provides a significant advantage over existing alternatives.
- The diversity of the eight 3D scenes may be insufficient for ensuring generalization to unseen scenes, potentially limiting the finetuned model's robustness.

Conclusion

The paper is well-written, and the research problem it addresses (inconsistencies in image editing) is both valid and important. The experiments also clearly demonstrate the benefits of the proposed dataset over using vanilla image editing alternatives. However, the absence of a direct comparison with other datasets weakens the claims. Given these considerations, I rate this paper marginally above the acceptance threshold.

---

### Official Review · Reviewer_LgKb · 2025-03-27
**Nighttime-daytime image translation review**

**Rating:** 6
**Confidence:** 4

**Review:**

Summary: The paper address the problem of generating daytime from night time images. The paper itself is hard due to the lack of paired data. The paper propose a synthetic data generation pipeline to create those synthetic pairs. First, the authors train a NeRF model on daytime images and then use 3D editing technique to transform render daytime view to night time. Those collected night time images are being used to finetune difffusion based models.

Strengths:
- The paper is clear and well-written.
- The proposed synthetic data generation makes sense and novel.

Weakness:
- The method lacks the comparisons with state-of-the-art image editing techniques. It is not clear if the proposed method is able to outperform those editing techniques.
- The method is tested on unseen data but the number of samples are quite limited.

---

### Official Review · Reviewer_KEZ1 · 2025-03-27
**Review paper number 1**

**Rating:** 6
**Confidence:** 4

**Review:**

Summary: This paper proposed the nighttime-to-daytime translation dataset, which focuses on geometry consistency. There are four high level steps to create this data sets. The day images are then used to create 3D scenes using NeRF. After that, Instruct-NeRF2NeRF and IP2P work to convert these scenes into nighttime styles using prompts including “Dark night” and “Moonlight.” Next, the rendering step retains only clear, geometry-consistent views and filters out low-quality renderings. Lastly, the final synthetic nighttime-daytime image pairs are used to fine-tune further models such as Stable Diffusion v1. 5 and Instruct-Pix2Pix, which enhance nighttime-to-daytime image translation. In the end, this dataset contained 1,812 image pairs.

Strengths:
+ This dataset helped the model improve significantly when finetuned on it.
+ The process to make the dataset is clear and valuable to the research community

Weakness:
+ This dataset only has 1,812 image pairs, which is still limited. I hope that the authors can increase this up to 5,000 image pairs
+ Because this dataset focuses on geometry consistency, I hope that the authors should do some 3d reconstruction model on this dataset to benchmark the geometry consistency property.

---

### Decision · Program_Chairs · 2025-03-30

**Decision:**

Accept (Poster)

**Comment:**

The paper received all scores as 6 (Borderline Accept). Overall, the paper was applauded for its interesting problem setup, clear writing, reasonable algorithm, and promising qualitative results. However, the paper was also criticized for missing comparison to state-of-the-art image editing techniques, particularly the ones finetuned on existing nighttime-to-daytime datasets. The dataset size was small, and 3D reconstruction experiments should be added to prove geometry consistency. The experiment on the out-of-distribution dataset, which was more critical to verifying the usefulness of the proposed method, was limited and had no quantitative number.

Despite the shortcomings, the Program Chairs agreed that the paper had some merits, with potentially valuable contributions to the research community. Hence, we agreed to accept the paper. The authors should check the reviewers' concerns and improve the paper in the camera-ready version, e.g., adding comparisons to state-of-the-art image editing techniques that are finetuned on existing nighttime-to-daytime datasets.